# A Neurosymbolic Approach to Counterfactual Fairness

**Xenia Heilmann**                                    XENIA.HEILMANN@UNI-MAINZ.DE
*Institute of Computer Science, Johannes Gutenberg University, Mainz, Germany*

**Chiara Manganini**                                    CHIARA.MANGANINI@UNIMI.IT
*Department of Philosophy, University of Milan, Milan, Italy*

**Mattia Cerrato**                                    MCERRATO@UNI-MAINZ.DE
*Institute of Computer Science, Johannes Gutenberg University, Mainz, Germany*

**Vaishak Belle**                                    VBELLE@ED.AC.UK
*School of Informatics, University of Edinburgh, Edinburgh, United Kingdom*
*Alan Turing Institute, London, United Kingdom*

**Editors:** Leilani H. Gilpin, Eleonora Giunchiglia, Pascal Hitzler, and Emile van Krieken

## Abstract

Integrating fairness into machine learning models has been an important consideration for the last decade. Here, neurosymbolic models offer a valuable opportunity, as they allow the specification of symbolic, logical constraints that are often guaranteed to be satisfied. However, research on neurosymbolic applications to algorithmic fairness is still in an early stage. With our work, we bridge this gap by integrating counterfactual fairness into the neurosymbolic framework of Logic Tensor Networks (LTN). We use LTN to express accuracy and counterfactual fairness constraints in first-order logic and employ them to achieve desirable levels of both performance and fairness at training time. Our approach is agnostic to the underlying causal model and data generation technique; as such, it may be easily integrated into existing pipelines that generate and extract counterfactual examples. We show, through concrete examples on three real-world datasets, that logical reasoning about counterfactual fairness has some important advantages, among which its intrinsic interpretability, and its flexibility in handling subgroup fairness. Compared to three recent methodologies in counterfactual fairness, our experiments show that a neurosymbolic, LTN-based approach attains better levels of counterfactual fairness.

## 1. Introduction

In the last decade, there has been a considerable amount of research on the topic of fairness in deep learning, as neural networks are increasingly used in critical contexts such as credit scoring, risk assessment of recidivism, and job recruitment. As of today, making these systems fairer is a complex and multi-faceted challenge. In this context, the idea of leveraging neurosymbolic approaches to tackle algorithmic unfairness has been largely underexplored so far. The potential for a good fit between these two research lines has been pointed out in the recent surveys by Gibaut et al. (2023) and Bhuyan et al. (2024). Neurosymbolic AI allows one to reason symbolically about the neural network's behaviour, by establishing a correspondence between its low-level information processing and high-level logical reasoning (Hitzler and Sarker, 2022; Sarker et al., 2021). As such, the approach shows many advantages for establishing trust in deep learning systems, by making models more interpretable and transparent (Gibaut et al., 2023).

To bridge this gap, we propose an in-processing method to train a counterfactually fair neural network by means of the neurosymbolic method of Logic Tensor Networks (LTN) proposed by Badreddine et al. (2022). Counterfactual fairness (CF) reframes the problem of algorithmic fairness in terms of causality, by asking the question: "Would I be treated in the same way, had my protected feature been different?". In short, it requires that, for a machine learning model to be fair, sensitive attributes (e.g., race, gender, etc.) have no causal influence on its outcomes. In the present work, we integrate CF into the neural network learning process, in the form of logical constraints. Furthermore, we show how to exploit symbolic reasoning after network training to better secure fairness for specific subgroups of sensitive groups. Lastly, we integrate a counterfactual knowledge extraction method into the LTN training process. We evaluate our method on three real-world datasets with binary as well as score-based predictions. We find that our method is able to take accurate decisions with minimal infractions in terms of counterfactual fairness, especially when subgroup fairness is considered. The main contributions of this work are the following. First, we push the state-of-the-art in neurosymbolic fairness approaches by showing how to integrate counterfactual fairness and subgroup counterfactual fairness into LTN. Secondly, we introduce a novel methodology to automatically extract fairness constraints from counterfactual explanations.

## 2. Related Work

**Counterfactual Fairness.** According to counterfactual fairness, a classifier treats individuals fairly if they would have received the same outcome, had their sensitive attribute been different. The computation of such a counterfactual outcome requires knowledge of the structural causal model[1] $\mathcal{M}$ underlying the data-generating process. Knowing $\mathcal{M}$, the computation of the counterfactual outcome corresponds to the intervention $\hat{Y}_{S \leftarrow s'}$ (Pearl and Mackenzie, 2018), which denotes the value of the predicted outcome $\hat{Y}$ – as determined by the structural equations of $\mathcal{M}$ – once sensitive feature $S$ has been set to $s'$. The original formalisation by Kusner et al. (2017) is given in probabilistic terms as the requirement that the probability distribution of the outcomes is the same in the actual world, where $S = s$, and in the counterfactual world, where $S = s'$. This must hold for any individual i.e., under any assignment of sensitive feature $S$ and non-sensitive features $A$ in the actual world:

$$P(\hat{Y}_{S \leftarrow s} = y | S = s, A = a) = P(\hat{Y}_{S \leftarrow s'} = y | S = s, A = a) \quad \forall y \in \hat{Y}, a \in A, s, s' \in S \quad (1)$$

Among the approaches to counterfactual fairness, the majority of the work proposes to enforce it by generating counterfactual data, and then use this data to enhance factual training data to input into a machine learning training pipeline (Javaloy et al., 2023; Zuo et al., 2023; Louizos et al., 2017; Kim et al., 2021; Lin et al., 2024; Xu et al., 2019; Kocaoglu et al., 2018; Yang et al., 2021). The main focus throughout these approaches lies on the

---

1. A structural causal model $\mathcal{M}$ is defined as the tuple $\langle U, V, F \rangle$, where $U$ is the set of exogenous variables whose values are determined by factors outside the model and are, therefore, taken "as given"; $V$ is the set of endogenous variables, whose values are ultimately determined by the exogenous variables; $F$ is the set of structural equations that determine the value of each endogenous variable, as a function of other endogenous and exogenous ones. The predicted outcome of the ML model $\hat{Y}$ is an endogenous variable. Let us denote the set of non-sensitive features as $A = (V \cup U) \setminus \{S, \hat{Y}\}$.

counterfactual generation process leaving aside modifications on the final predictor itself. Differently, Grari et al. (2023) claim that additionally integrating counterfactual fairness objectives into the loss function of the machine learning pipeline contributes to counterfactually fairer predictions. Our proposal builds on the latter suggestion and develops the idea of a neurosymbolic approach in which the requirements of counterfactual fairness are expressed logically and injected at training time.

**Logic Tensor Networks.** The LTN framework integrates a fully differentiable first-order logic $\mathcal{L}$ with a fuzzy semantics. Its signature contains a set of constants $\mathcal{C}$, function symbols $\mathcal{F}$, variables $\mathcal{X}$, and predicate symbols $\mathcal{P}$. Symbols are interpreted by their grounding $\mathcal{G}$ onto real numbers. Every object denoted by a constant, variable, or term is grounded onto a tensor of real number; function symbols are grounded as $n$-ary functions that map $n$ vectors of real numbers to one vector of real numbers; predicates are grounded as functions that map onto the interval $[0, 1]$ representing their degree of truth. Connectives are interpreted according to fuzzy logic semantics, while the universal quantification is defined as the generalised p-mean (Badreddine et al., 2022; Wagner and d'Avila Garcez, 2021).

**Fairness Through Neurosymbolic Methods.** The paper by Wagner and d'Avila Garcez (2021) has recently inaugurated a line of research that combines algorithmic fairness with neurosymbolic aspects. Here, the authors propose a general method for instilling fairness constraints into deep network classifiers. They apply the LTN framework and inject these fairness constraints as logically expressed axioms. Then, the learning process feeds back until these are satisfied. Their work focuses on the group fairness metrics of demographic parity and disparate impact for which the reported experiments reveal that fairness with respect to these metrics is achieved without sacrificing accuracy. Furthermore, it is experimentally shown by Greco et al. (2023) that the effectiveness of LTN for securing fairness is highly dependent on the semantic interpretations chosen, and that the optimal combination of them yields results in line with previous non-neurosymbolic approaches to group fairness.

**Counterfactual Explanations.** One active line of research in fair machine learning explores the possibility of using XAI methods to discover and even impose fairness (Deck et al., 2024). In the pipeline by Wagner and d'Avila Garcez (2021), the SHAP explainability method (Lundberg and Lee, 2017) is used, but plays no active role, as it is only employed to isolate problematic imbalances and subsequently check the efficacy of their fairness constraints in their mitigation. In contrast, in our pipeline we can exploit counterfactual explanationsfor the *automatic generation and injection* of *ad hoc* fairness constraints into the network. The counterfactual explanation of a negatively predicted point is defined as the (set of) nearest feature combination(s) obtaining a favourable prediction. Our method is inspired by the one that Goethals et al. (2024) have introduced to detect significant differences in the distribution of counterfactual explanations between sensitive groups. As an example, they show that in the Adult dataset women are more frequently returned *marital-status* as a counterfactual explanation, in comparison to men. The feature *marital-status* is not often considered sensitive (like gender), but still it is generally thought of as *immutable*, i.e., a feature that individuals cannot or do not want to act upon. Since it is problematic to suggest that an individual should change immutable features of this type to obtain a positive outcome, we maintain that counterfactual explanations involving immutable features are undesirable. For this reason, in Section 3.3, we develop a method to smooth out possible imbalances in undesirable counterfactual explanations between sensitive groups.

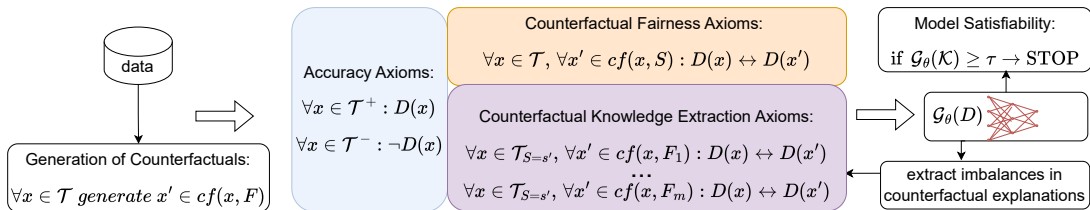

Figure 1: Overview of our pipeline for binary predictions. Here, $S$ denotes the set of sensitive attributes, $F_i$ a feature, $\mathcal{T}$ the dataset and $\mathcal{D}$ the prediction model.

## 3. Method

The goal of our pipeline is to enforce counterfactual fairness, while preserving accuracy of predictions, and additionally disincentivising undesirable counterfactual explanations that suggest to intervene on immutable features to achieve a favourable outcome. Specifically, we define certain data columns as *immutable* if they are either sensitive or particularly challenging for individuals to act upon. We give a mapping of feature columns into immutable, sensitive and actionable in Appendix A. We attain these three goals by integrating adequate axioms into the training process of the LTN framework.

An overview of the pipeline for datasets with binary outcomes can be found in Figure 1. As a first pre-processing step, we approximate counterfactual examples for all data points $x$ in the data $\mathcal{T}$. Here, any counterfactual generation method can be applied. Secondly, we introduce axioms to ensure accuracy, followed by the axioms enforcing counterfactual fairness. Finally, we add axioms from our counterfactual knowledge extraction method, which disincentivise counterfactual explanations intervening on an immutable feature. We then train a model, which can *post-hoc* be queried for imbalances between sensitive subgroups. The result can be fed back into the training pipeline by adding additional axioms and retraining our model until model satisfiability is reached. Our pipeline is capable to handle, with different sets of axioms, both binary predictions and score-based ones.

### 3.1. Accuracy Axioms

The first axioms we add to the training pipeline ensure the accuracy of the model predictions. Here, for binary predictions, we adapt the axioms for predictive performance by Wagner and d'Avila Garcez (2021). Let $D$ denote our classifier, $\mathcal{T}$ our dataset and let $x \in \mathcal{T}$ hold. Furthermore, let $\mathcal{T}^+$ be the set of data points with a positive outcome as ground truth and $\mathcal{T}^-$ the data points with a negative outcome as ground truth. Then, we can state the following axioms:

$$\forall x \in \mathcal{T}^+ : \quad D(x) \tag{A1}$$

$$\forall x \in \mathcal{T}^- : \neg D(x) \tag{A2}$$

For a score based prediction, our axioms have to take into account that predictions and ground truth are close to each other. We therefore define a predicate for the equality

$Eq(\hat{y}, y) = 1/(1 + 0.5\sum_j(\hat{y}_j - y_j)^2)$, where $\hat{y}$ denotes the predicted score of the data points $x \in \mathcal{T}$. With $y$ as ground truth score, we then have the axiom optimizing the predictive performance for score based settings:

$$\forall x \in \mathcal{T} : Eq(D(x), y) \tag{A3}$$

### 3.2. Counterfactual Fairness Axioms

By adding the axioms, we want that a data point and its counterfactual with respect to the sensitive attribute $S$ receive the same outcome. Let $x' \in cf(x, S)$ denote the set of generated counterfactuals of $x$ with respect to the sensitive feature $S$. Intuitively, we have that for counterfactual fairness

$$\forall x \in \mathcal{T}, \forall x' \in cf(x, S) : D(x) \leftrightarrow D(x') \tag{A4}$$

should hold. Adding Axiom A4 will guarantee us an overall counterfactually fairer model as it reformulates the original definition of counterfactual fairness expressed in (1), as a first-order logic constraint. Note that, as before, we need to modify Axiom A4 for score-based prediction by reformulating $D(x) \leftrightarrow D(x')$ to $Eq(D(x), D(x'))$ (as defined above) to check for closeness of predicted scores. This holds for all further axioms. We now go one step further, showing how to integrate counterfactual fairness axioms for subgroups (or "subgroup counterfactual fairness"). The rationale here is that Axiom A4 does not capture whether the model is less fair to a sensitive subgroup in the dataset than to another subgroup. We therefore refine our axioms with respect to subgroups $C_1, \ldots, C_n$ as follows:

$$\forall x \in \mathcal{T}_{C_1}, \forall x' \in cf(x, S) : D(x) \leftrightarrow D(x') \tag{A4$_1$}$$

$$\ldots$$

$$\forall x \in \mathcal{T}_{C_n}, \forall x' \in cf(x, S) : D(x) \leftrightarrow D(x') \tag{A4$_n$}$$

In a simple setting, the data could be divided into subgroups with different sensitive values. Yet, also more refined subgroups are supported, i.e. further partitioning the sensitive groups (e.g., females and males) into subgroups based on other features (e.g., age). These subgroups can be designed to partition the entire dataset (for instance, "young females", "elderly females", "young males", and "elderly males"), or to isolate a specific subset of interest within the sensitive group, for which we want to impose the fairness constraint. This is especially interesting in real-world scenarios where counterfactual fairness might not be relevant for all subgroups of the sensitive feature, but only for some of them. For instance, a financial institute might want to evaluate the counterfactual fairness w.r.t. gender of a loan that can be granted to young people only, or certain professionals only (e.g., teachers). In this case, we would apply axiom A4$_n$ for $\forall x \in \mathcal{T}_{young}$ or $\forall x \in \mathcal{T}_{teachers}$. This setup makes our approach adaptable to many applications in which subgroup counterfactual fairness is to be attained.

### 3.3. Counterfactual Knowledge Extraction Axioms

We want our pipeline to be able to detect an imbalance in the frequency of undesirable counterfactual explanations between sensitive groups and automatically generate *ad hoc*

axioms to mitigate such an imbalance. To this end, we generate counterfactual explanations of negatively predicted data points.[2] We then compare the frequencies of counterfactual explanations across groups by aggregating the data points on the basis of the sensitive attribute. Thus, we obtain the difference of frequencies for undesirable explanations. This score provides an analyst with a way to extract valuable information on which specific counterfactual explanations are not desirable, and for which sensitive class. Let us denote these explanations by $(s', F_1), \ldots, (s', F_m)$ and $\mathcal{T}_{S=s'}$ the dataset for which the sensitive attribute is $s'$. We want for a data point $x$ that its counterfactual explanations $x' \in cf(x, F_i)$ with respect to feature $F_i$, receives the same outcome as $x$, indicating that feature $F_i$ is not relevant to the outcome of the prediction. This can be modelled by the following axioms:

$$\forall x \in \mathcal{T}_{S=s'}, \, x' \in cf(x, F_1) : D(x) \leftrightarrow D(x') \tag{A5$_1$}$$

$$\ldots$$

$$\forall x \in \mathcal{T}_{S=s'}, \, x' \in cf(x, F_m) : D(x) \leftrightarrow D(x') \tag{A5$_m$}$$

While for counterfactual fairness axioms we add all axioms at the same time in the training pipeline, these axioms are added iteratively for better model surveillance and oversee their individual influence to counterfactual fairness. Furthermore, a human-in-the-loop may be integrated in this part of the pipeline to assess which constraints are desirable to be integrated as axioms.

## 4. Experiments

We conducted experiments to showcase that integrating accuracy, CF and axioms from counterfactual knowledge extraction is beneficial for training counterfactually fairer models.[3] Specifically, we address the following research questions:

(Q1) How does our method improve overall and subgroup counterfactual fairness?
(Q2) How does our method compare to other approaches in terms of fairness and accuracy?
(Q3) Can counterfactual knowledge extraction be exploited to learn effective axioms?

**General Setup.** We conduct experiments on the **Adult, COMPAS, COMPAS(age)**, and **Lawschool** datasets. Dataset details can be found in Appendix A. Our function for approximating counterfactual examples $x' \in cf(x, F)$ is implemented via causal normalizing flows (Javaloy et al., 2023) (for details, see Appendix B). Yet, we stress that our method does not train to generate counterfactual examples but only requires them as input, and may be employed in conjunction with any counterfactual generation methodology. These generation methods can be applied in a pre-processing step and the generated counterfactuals can then serve as input into our pipeline. As predicate for prediction in LTN, we train a multi-layer perceptron (MLP) with two layers of 100 and 50 neurons trained with the Adam optimizer with learning rate 0.1. We report averaged results over a 5-fold cross-validation. For LTN, we use Reichenbach implication and $p = 1$ as universal quantifier's exponent (Badreddine

---

2. Limiting ourselves to those interventions on one feature only, which result in a change of the original prediction. Our interventions set the feature to the most frequent (max 10) values that are at least present in 1% of the training set. For continuous features, we take the percentiles.

3. Our code can be found at https://github.com/xheilmann/CounterfactualFair_LTN

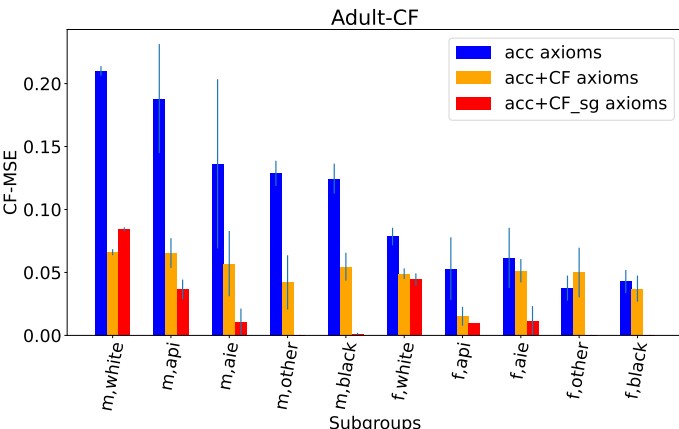

Figure 2: CF-MSE for the Adult dataset in three different axiom settings for each subgroup in (*gender, race*). Male corresponds to *m*, female to *f*, and *asian-pac-islander, american-indian-eskimo* are abbreviated with *api* and *aie*, respectively.

et al., 2022). In this section, we evaluate our method for equally weighted axioms. We report our results with imbalanced weight settings in Appendix D.4.

### 4.1. Q1: Fairness and subgroup axioms improve counterfactual fairness

**Setup.** To show the effectiveness of our method, we evaluate three different axiom settings. As a baseline, we only apply the accuracy axioms (Equation A1-A2 or A3) to our pipeline (acc axioms). Next, we integrate the CF axiom (Equation A4) in addition to the accuracy axioms (acc+CF axioms). Lastly, we evaluate on the combination of subgroup axioms (Equation $A4_1$-$A4_n$) and accuracy axioms (acc+CF_sg axioms). All settings employing fairness axioms, pre-train an LTN for 1500 epochs on the accuracy axioms, then add the CF axioms. We measure the degree of fairness in a model's decision by computing the mean squared error (MSE) between the predictions made for factual data points and their counterfactuals: $\frac{1}{n}\sum_{x\in\mathcal{T},x'\in cf(x,S)}|D(x) - D(x')|^2$, which we will call CF-MSE for clarity (Grari et al., 2023). Lower values of CF-MSE indicate a counterfactually fairer model.

**Results.** Results for **Adult** are displayed in Figure 2. There, one can see that applying the CF axioms strongly increases fairness for the majority of subgroups. The greatest improvement in CF-MSE can thereby be seen for the largest subgroup, namely *white males*, whereas for the female subgroups fairness only improves slightly and even gets worse for *females* in the "other" ethnic subgroup. However, integration of the subgroup axioms into the training objective mostly prevents this phenomenon. Overall, CF improves for all subgroups upon the accuracy-only baseline; for all subgroups but white males, the CF-MSE is again improved by adding subgroup CF axioms. The same holds for the **COMPAS** dataset. In the top row plots of Figure 3, we show that CF axioms as well as subgroup CF axioms improve CF-MSE over all subgroups. We give complete numerical results for our

Table 1: Comparison of our pipeline (three different axiom settings) with current baselines evaluated on CNF approximated counterfactuals in terms of accuracy, CF-MSE and worst subgroup CF-MSE (sg) as average of 5 runs. Row-wise best results are in bold.

| dataset | metric | LTN (our pipeline) | | | CNF | GAN | DCEVAE |
|---|---|---|---|---|---|---|---|
| | | acc | acc+CF | acc+ CF_sg | | | |
| Adult | accuracy ↑ | 0.782±0.006 | 0.758±0.01 | 0.812±0.001 | 0.825±0.001 | 0.777±0.005 | **0.831±0.002** |
| | CF-MSE ↓ | 0.160±0.006 | 0.065±0.002 | **0.055±0.001** | 0.074±0.009 | 0.216±0.011 | 0.109±0.009 |
| | CF-MSE (sg)↓ | 0.210±0.004 | **0.066±0.002** | 0.084±0.044 | 0.113±0.000 | 0.263±0.018 | 0.291±0.013 |
| COMPAS | accuracy↑ | 0.671±0.010 | 0.675±0.003 | 0.651±0.013 | 0.661±0.003 | **0.685±0.002** | 0.665±0.009 |
| | CF-MSE↓ | 0.156±0.037 | 0.047±0.004 | **0.045±0.006** | 0.072±0.010 | 0.107±0.012 | 0.188±0.051 |
| | CF-MSE(sg)↓ | 0.208±0.094 | 0.045±0.002 | **0.043±0.092** | 0.086±0.010 | 0.110±0.012 | 0.194±0.061 |
| COMPAS (age) | accuracy↑ | 0.658±0.013 | 0.654±0.016 | 0.658±0.012 | 0.667±0.002 | **0.675±0.001** | 0.651±0.008 |
| | CF-MSE↓ | 0.254±0.032 | 0.079±0.016 | **0.075±0.016** | 0.094±0.003 | 0.171±0.010 | 0.244±0.044 |
| | CF-MSE(sg)↓ | 0.428±0.048 | **0.131±0.021** | 0.142±0.057 | 0.181±0.006 | 0.204±0.013 | 0.342±0.074 |
| Lawschool | MSE↓ | 0.767±0.013 | 0.782±0.002 | 0.796±0.013 | 0.771±0.008 | 0.906±0.000 | **0.754±0.024** |
| | CF-MSE↓ | 0.096±0.028 | 0.003±0.000 | **0.001±0.000** | 0.012±0.002 | 0.227±0.013 | 0.210±0.042 |
| | CF-MSE(sg)↓ | 0.358±0.021 | 0.011±0.002 | **0.001±0.002** | 0.014±0.003 | 0.272±0.025 | 0.251±0.046 |

LTN pipeline on all considered datasets in Table 1. Therein, we include average CF-MSE and worst-subgroup CF-MSE for all datasets and methods considered. The clear trend is that the average CF-MSE across groups improves when applying subgroup fairness axioms for all datasets. In COMPAS (age) and Adult, we even observe small increases in terms of accuracy. For **Lawschool**, subgroup CF axioms efficiently reduce the worst subgroup CF-MSE to 0.001 (also see Appendix D.1). We conclude, in terms of **Q1**, that our methodology has a positive impact in terms of fairness, esp. when subgroups are actively considered.

### 4.2. Q2: Strong increase in fairness

**Setup.** We choose the following approaches as baselines: DCEVAE (Kim et al., 2021), causal normalizing flows (CNF) (Javaloy et al., 2023) and a GAN-based method (Grari et al., 2023) (for implementation details, see Appendix C). For DCEVAE and CNF, we trained an MLP with the same hyper-parameters as our pipeline on the combined set of generated counterfactual and factual data points. Note that these methodologies, differently from ours, generate counterfactual examples themselves. This also means that they are not agnostic to the specific generated data points. To keep the comparison as fair as possible, we test all methodologies on the same counterfactual data, which we generate using a causal normalising flow model (Javaloy et al., 2023). Results on differently generated counterfactuals are given in Appendix D.3. We then report CF-MSE, accuracy (Adult, COMPAS), MSE (Lawschool) as well as worst subgroup CF-MSE for all methodologies and datasets. Here, worst subgroup CF-MSE denotes the worst CF-MSE value for the subgroups we evaluate our method on.

**Results.** We provide a complete comparison in Table 1. Regarding the comparison with CNF, results show that our pipeline, which adds counterfactual fairness constraints during training, significantly improves CF compared to only pre-processing for fairness as done by CNF. However, except for **COMPAS**, this results in a decrease in accuracy compared to CNF. For the GAN-based method, we can see improved accuracy for COMPAS but

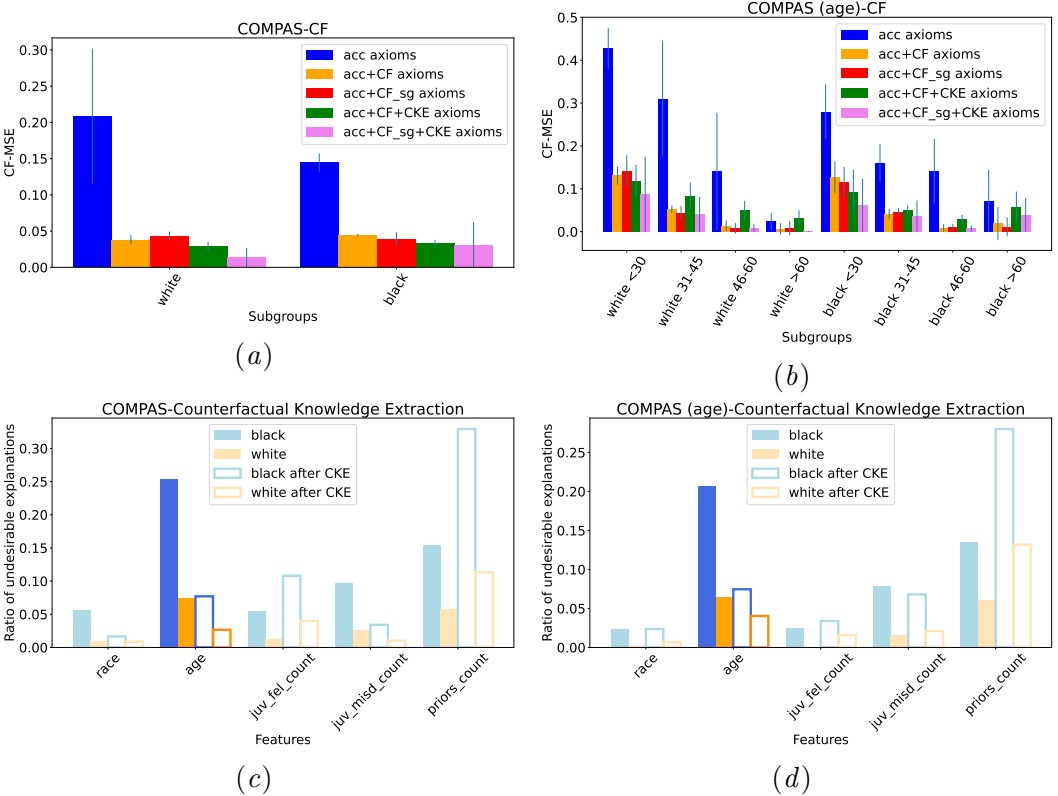

Figure 3: Top: Development of CF-MSE for our pipeline for 5 different axioms settings for both COMPAS datasets. Bottom: Ratio of undesirable explanations for each sensitive group before and after applying a CKE axiom for *(black, age)*.

worse overall CF as well as worse subgroup CF in comparison to our method. Similarly, DCEVAE has strong results in terms of accuracy and MSE, but struggles in achieving counterfactually fair results. Here, our results differ significantly from the ones reported by the original authors (Kim et al., 2021). Our empirical, if anecdotal, experience with DCEVAE is that it struggles to converge to an accurate result, and even that comes at the expense of fairness. We elaborate on these reproducibility challenges in Appendix C.

### 4.3. Q3: Counterfactual knowledge extraction learns effective fairness axioms

**Setup.** Our pipeline, as described in Section 3.3, integrates counterfactual knowledge extraction (CKE henceforth). To summarise, CKE detects imbalances across sensitive groups in the frequency of undesirable counterfactual explanations, learning new training axioms to reduce them. To generate the counterfactual of data point $x$ with respect to a generic immutable feature $F$ (denoted by $cf(x, F)$) we use the method of causal normalizing flows described in Appendix B. We define imbalance as a difference in frequencies of at least 0.1 for COMPAS, and 0.01 for Adult and Lawschool.

**Results.** We applied the extracted axioms both on top of models trained with the CF axiom (acc+CF+CKE) and with subgroup CF axioms (acc+CF_sg+CKE). Results for **COMPAS** are reported in Figure 3 as well as in Table 2 of Appendix D.2 . Here, the CKE deduced a strong imbalance for *age* for the *black* subgroups. After enforcing that *age* be irrelevant for decisions made in the *black* group, the imbalance drops below the threshold. However, deincentivising *age* as counterfactual explanation results in the imbalance widening for other attributes – especially *priors_count* (Figures 3(*c*) and 3(*d*)). In the same figure, we observe that CF increases for both ethnicity subgroups even though axioms are only added for the *black* subgroup. A further increase of CF is achieved in the combination of subgroup CF axioms and the CKE axiom (Figures 3(*a*) and 3(*b*)). For **Lawschool**, as the counterfactual knowledge extraction works on binary predictions, we map the best 40% of all scores to a positive outcome. As a result, for Lawschool for one out of five runs *race* was detected as undesirable explanation for the *female* subgroup when CKE was evaluated after only training with the accuracy axioms. Yet, when CF axioms were added *race* was not detected as undesirable explanation anymore. Therefore, we conclude that CF axioms in this setting already eliminate undesirable explanations efficiently enough. For **Adult**, we refer to Table 3 in Appendix D.2. Therein, *race, marital-status, native-country* were interestingly detected as undesirable explanations for *males*. Here, for each detection a CKE axiom was added subsequently after 500 additional training epochs (ordered from highest imbalance to smallest imbalance), after which we each checked the axiom's impact on accuracy, CF-MSE and worst subgroup CF-MSE. Due to the axiom ordering by imbalance level, we have different sequences in which axioms are added for each run. Overall we found that accuracy stays stable throughout the CKE process. However, while CF is improved, subgroup CF gets worse with each additional CKE axiom after the first. It is left for further research how to establish scalability of the method beyond a single CKE axiom. Our takeaway on the CKE technique (**Q3**) is that it is indeed able to learn beneficial axioms that reduce specific unfairness patterns for certain subgroups and feature combinations. It is also possible to employ a similar principle to extract counterfactual examples for algorithmic recourse. We present a preliminary investigation of this technique in Appendix D.5.

## 5. Conclusion and Future Work

To conclude, we have shown how to integrate the individual-based notion of counterfactual fairness into an LTN training pipeline. We proposed axioms for this integration and refined the axioms to subgroups to achieve higher counterfactual fairness in these. Furthermore, we integrated counterfactual knowledge extraction into our pipeline with subsequent axiom extraction to discourage undesirable counterfactual explanations. Our pipeline improves counterfactual fairness and decreases the discrepancy between subgroups w.r.t. the unfair baseline. This paper and the previous work we relate to suggest that the neurosymbolic approach to fairness is promising. For one thing, it allows us to explicitly codify fairness axioms, but also potentially balance different axioms depending on the trade-offs/constraints for the application at hand. On the other hand, various methods are currently applied to check that the prediction model actually satisfies these constraints after training. Thus, we expect that there will be more in this line of work in the future, which makes the need for a robust system all the more pertinent.

## Acknowledgments

XH and MC were supported by the "TOPML: Trading Off Non-Functional Properties of Machine Learning" project funded by Carl Zeiss Foundation, grant number P2021-02-014. Author VB was supported by a Royal Society University Research Fellowship.

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

## Appendix A. Datasets

We conduct experiments on the **Adult** dataset (Becker and Kohavi, 1996), with *gender* as our sensitive attribute. As subgroups, we take each attribute combination of (*gender, race*). As immutable features we identify *marital-status, relationship, race* and *native-country*. The test set contains around 10K data points and their corresponding counterfactuals.

Furthermore, we apply our method on the **COMPAS** dataset (Angwin et al., 2016) with *race* as the sensitive attribute. Here, we evaluate on and add subgroup fairness axioms for *race* and each attribute combination of (*race, age*). For the latter, we group the *age* attribute into four categories, namely, under 30, 31-45, 46-60, and older than 60 years (**COMPAS(age)** in the following). As immutable feature we have *age*. In our test set we have on average 1230 data points and their counterfactuals.

As a third dataset, we employ the **Lawschool** dataset (Wightman, 1998) with *gender* as the sensitive attribute. We want to stress that Kusner et al. (2017) show that this dataset is counterfactually fair with respect to gender. Here, we evaluate how adding our (subgroup) counterfactual fairness axioms improve subgroup fairness for all combinations of (*gender, race*). We have *race* as immutable feature. We evaluate on a test set containing 4359 data points.

## Appendix B. Estimation of Counterfactuals

To estimate the counterfactuals of our (factual) observations, we applied the methods of Causal Normalizing Flows (CNF) (Javaloy et al., 2023) due to their relative simplicity of implementation with respect to the other two methods (GAN and DCEVAE, see Appendix C). CNF are causal generative models that leverage on the deep-learning method of normalizing flows to accurately and efficiently approximate the structural causal model $\mathcal{M}$ – as defined in Section 2 – of a data-generating process. The approximation is carried out on the basis of (factual) observations and the causal graph induced by $\mathcal{M}$. The causal graph induced by $\mathcal{M}$ is a directed acyclic graph whose nodes are labelled by the endogenous and exogenous variables of $\mathcal{M}$, and where each directed edge from node $a$ to node $b$ indicates that the latter *depends* on the former. The exogenous variables correspond to the roots of the graph. Unlike $\mathcal{M}$, the causal graph induced by it is in many cases obtainable through domain knowledge, as it is a description of the causal dependencies of $\mathcal{M}$, without specifying its structural equations. We extended the original code[4] to generate the counterfactuals with respect to *all* the attributes, rather than just for the sensitive one. This is necessary for the generation of the counterfactual knowledge extraction (CKE) axioms described in Section 3.3. For the interventions, we set the feature value to the most frequent values that are at least present in 1% of the dataset, up to a maximum of 10 values. For continuous features, we took their percentiles.

For generating counterfactuals, we trained the CNF on the partial[5] causal graphs by Zhang et al. (2016) for Adult, by Russell et al. (2017) for COMPAS, and by Kusner et al. (2017) for Lawschool. For COMPAS and Lawschool we provide the partial causal graphs

---

4. Available at https://github.com/psanch21/causal-flows
5. Following Javaloy et al. (2023), we do not model the causal dependencies between the predictors and the target variable.

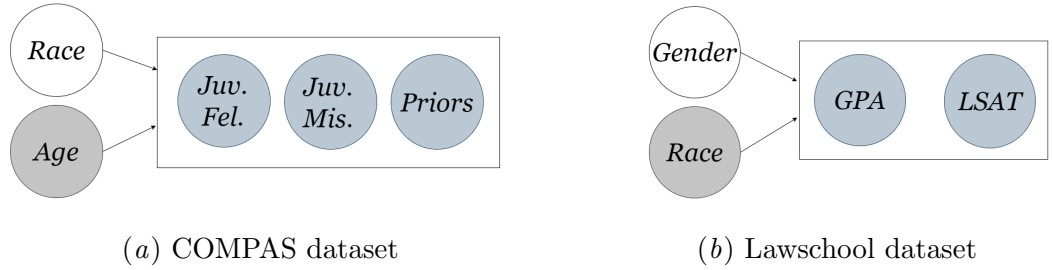

$(a)$ COMPAS dataset $\qquad$ $(b)$ Lawschool dataset

Figure 4: Partial causal graphs for the COMPAS (a) and Lawschool (b) datasets. The arrows connecting nodes and rectangles indicate that the node is connected to *every* node inside the rectangle. White nodes denote immutable sensitive features, grey nodes immutable non-sensitive features, and blue nodes actionable features.

in Figure 4. For training we kept the same hyperparameters as Javaloy et al. (2023), for all three datasets: 1000 epochs, batch size of 256, and inner dimension of $[32, 32, 32]$.

## Appendix C. Baselines

In this section, we provide more information on each of the three baselines our pipeline is compared to: GAN-based method, DCEVAE and CNF. Also, we report hyperparameter settings and adaptions made for the comparison. All the following methodologies, differently from ours, generate counterfactual examples themselves. Our approach, however, is agnostic to the underlying counterfactual generation technique and may be easily integrated in existing pipelines that generate and extract counterfactual examples. This presents a challenge in terms of comparison, as these methods will tend to perform better on the set of counterfactuals that they themselves generated compared to other methodologies. Hence, we provide an evaluation of each baseline on a test set of the counterfactuals (approximated by CNF) we input into our pipeline (results in the main paper) as well as a study on how our method performs when we input the counterfactuals generated by the GAN method (Appendix D.3). Furthermore, for DCEVAE and CNF, we train an MLP with the same hyperparameters as the underlying MLP in our method on the complete set of counterfactual and original data points. This is not to be confused with other proposed settings in literature (Javaloy et al., 2023), where predictors are sometimes trained in an *unaware* setting, which means that sensitive attributes are left out during training or only trained on non-descendent variables of the sensitive attribute.

**GAN-based method**. Grari et al. (2023) introduce a Generative Adversarial Model (GAN) approach for counterfactual inference and learning a counterfactually fair predictive model. For counterfactual inference, they propose a neural network encoder which generates a counterfactual from input $X$ (original data point), $Y$ and sensitive attribute $S$ and a decoder which tries to reconstruct original $Y$ and $X$ from the generated data point and $S$. The adversarial network tries to infer $S$ in this setting. For the counterfactual predictive

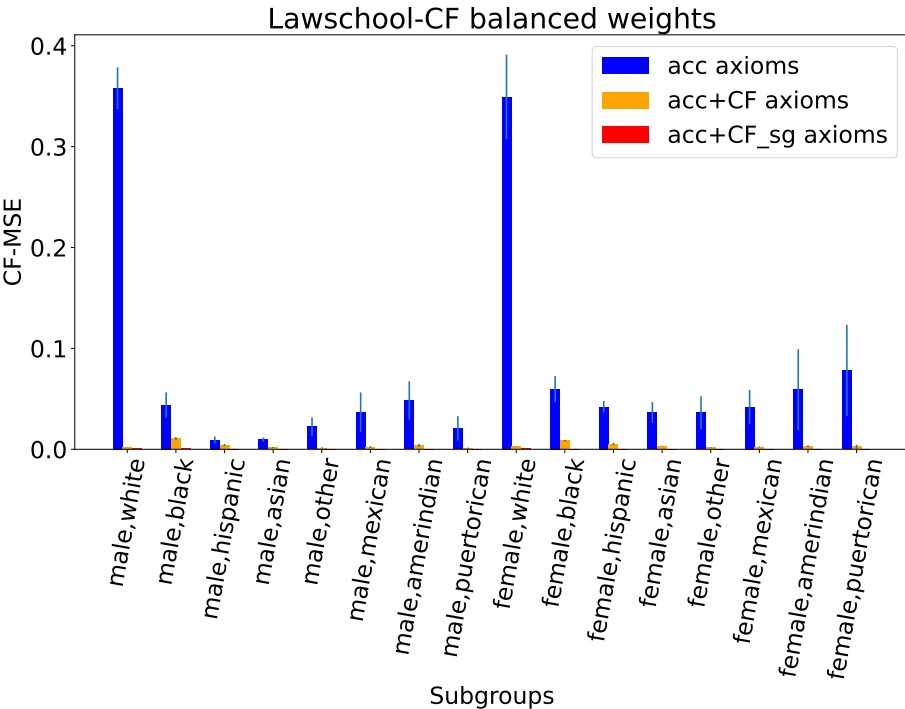

Figure 5: CF-MSE for Lawschool in three different axiom settings for each subgroup in (*gender, race*).

model, they add an additional term for penalizing counterfactual unfairness to their loss function and extend this method to continuous features.

We ran the available code[6] for 100 epochs for counterfactual inference and 1000 epochs for training a counterfactual fair predictor with learning rate 0.0001. For batch size we evaluated $[256, 512, 2048]$. Results are shown for 512 for Lawschool and COMPAS and 2048 for Adult. All other hyperparamters were set as given in the code.

**DCEVAE**. The Disentangled Causal Effect Variational AutoEncoder (DCEVAE) was proposed by Kim et al. (2021) as an extension to existing methodologies in fair variational optimisation. The main improvement put forward by the authors is the development of a ELBO-like objective for a causal graph in which variables that descend from sensitive attributes are kept separate from other covariates. The model then seeks to disentangle the VAE representations to separate the effects of the two sets of features. Among other applications, the authors test the counterfactual effect of applying their method to the Adult dataset.

In terms of integration into our experimental analysis, we started from the public code release by Kim et al. (2021) available at https://github.com/aailabkaist/DCEVAE. However, we noticed that the main PyTorch backprop code consistently gave a tensor version

---

6. From https://github.com/fairml-research/Counterfactual_Fairness

Table 2: Comparison of our proposed pipeline in five different axiom settings in terms of accuracy, CF-MSE and worst subgroup CF-MSE (sg) as average of 5 runs for both COMPAS datasets. CKE adds an axiom for *(black, age)*.

| dataset | metric | acc | acc+CF | acc+CF_sg | acc+CF+CKE | acc+CF_sg+CKE |
|---------|--------|-----|--------|-----------|------------|---------------|
| COMPAS | accuracy↑ | 0.671±0.010 | **0.675±0.003** | 0.651±0.013 | 0.645±0.008 | 0.633±0.022 |
| | CF-MSE ↓ | 0.156±0.037 | 0.047±0.004 | 0.045±0.006 | 0.032±0.001 | **0.025±0.006** |
| | CF-MSE(sg) ↓ | 0.208±0.094 | 0.045±0.002 | 0.043±0.092 | 0.033±0.005 | **0.031±0.031** |
| COMPAS | accuracy ↑ | **0.658±0.013** | 0.654±0.016 | 0.658±0.012 | 0.636±0.017 | 0.649±0.009 |
| (age) | CF-MSE ↓ | 0.254±0.032 | 0.079±0.016 | 0.075±0.016 | 0.077±0.027 | **0.047±0.016** |
| | CF-MSE(sg)↓ | 0.428±0.048 | 0.131±0.021 | 0.142±0.057 | 0.118±0.038 | **0.088±0.088** |

mismatch error. Thus, we modified the backprop loop by slightly changing the parameter update logic. We note that other authors that sought to reproduce the results from Kim et al. (2021) relied on the same bugfix.[7]

For hyperparameters we tested $[100, 250, 500]$ as training epochs and $[0.001, 0.0001]$ as learning rate as well as $[512, 1024]$ as batch size for all three datasets. We reported best results (100 epochs, 0.0001 learning rate, 1024 batch size) averaged over five runs.

**CNF**. We provide details on how we applied Causal Normalizing Flows (Javaloy et al., 2023) to approximate counterfactuals in Appendix B. Provided these counterfactuals, we train an MLP with the same parameters as for our methology on the combined dataset of counterfactuals and original data points. In terms of comparison, this baseline is the closest to our pipeline, as the same counterfactual generation method is applied. Yet, training is done differently as our method integrates counterfactual fairness constraints directly into the training pipeline and does not only take the generated counterfactual as input.

## Appendix D. Additional Experimental Results

We include here some additional experiments for Section 4. Furthermore we provide results to show that our method is agnostic to the input counterfactuals, how different weight settings influence the outcome and what additional knowledge extraction is possible for our method. We ran all experiments on a computer with specification Ubuntu 22.04.1 LTS, 64 GB RAM and Ryzen Threadripper 1920X 12-Core Processor as CPU. Running times ranged from 1 minute to 1.5 hours for the largest dataset.

### D.1. Additional Results Section 4.1

For Lawschool, in Figure 5 we show how CF-MSE changes for the balanced weight setting. As already mentioned in Section 4.1, we can see a huge improvement of CF for this dataset when adding CF axioms and an even stronger improvement when adding CF subgroup axioms. Therefore, even though when regarding gender as sensitive subgroup for Lawschool, we can provide counterfactual fairness for the sensitive intersectional subgroups $(gender, race)$.

---

7. For details, we refer to the `train.py` script on both the original repository, given above, and the following repository https://github.com/osu-srml/CF_Representation_Learning/blob/master/DCEVAE/train.py

Table 3: Impact for Adult of continuously adding CKE axioms on accuracy, CF-MSE and worst subgroup CF-MSE (sg). The CKE axioms are iteratively added in the order in which they appear in the table from left to right. Results are for one run, as the order of axioms varies across runs.

| metric | LTN(acc) | LTN(acc+CF) | LTN (acc+CF+CKE) | | |
| --- | --- | --- | --- | --- | --- |
| | | | *(race,male)* | *(mar.-status,male)* | *(nat.-country,male)* |
| accuracy ↑ | 0.778 | 0.755 | 0.757 | 0.755 | 0.755 |
| CF-MSE ↓ | 0.183 | 0.063 | 0.052 | 0.058 | 0.050 |
| CF-MSE (sg) ↓ | 0.224 | 0.073 | 0.070 | 0.100 | 0.139 |

## D.2. Additional Results for Section 4.3

In Table 2, we see the aggregated results for Figure 3. CF-MSE is greatly reduced overall but also for subgroups when adding CKE axioms to disincentivise *age* for the *black* subgroup. Yet, we see a trade-off between improved CF-MSE and a loss in accuracy when applying additional CKE axioms.

In Table 3 we show for Adult how subsequently adding the detected CFK axioms influences the results. The table shows that the accuracy stays stable throughout the entire CKE process. However, while CF is improved, subgroup CF gets worse with each additional CFK axiom after the first. This trend is also visible in the other runs. It is left for further research if only applying single CKE axioms or a specific addition order levels out these trade-offs.

## D.3. Applying our method to other counterfactuals

As stressed before, unlike ours, all methodologies we compare to generate counterfactual examples themselves. Our method relies only on a set of counterfactuals given as input, so that it is agnostic to the underlying counterfactual generation technique. However, during comparison of the different methods we faced the challenge that just a comparison of methods without taking the generated counterfactuals into account is not appropriate for our method. We therefore firstly compared each baseline on a common test set of the counterfactuals (approximated by CNF) we input into our pipeline. Secondly, we took the counterfactuals generated by the GAN-based method as input into our method and compared it to the GAN pipeline. For this comparison, we had to modify the GAN-based method, as in the original version CF-MSE and accuracy is calculated on different data encodings which was not possible as input into our pipeline. In Table 4 the results show better values for CF-MSE when training with our method. For COMPAS and COMPAS(age) this results in a decreased accuracy, compared to the GAN-based method. However, for the Adult and Lawschool dataset accuracy and MSE is improved upon the GAN method. Altogether, these results show that our method is applicable to counterfactuals generated with different methods than with CNF. Also, for these counterfactuals or method shows improved results, specifically for CF-MSE, when compared to the original generation method.

Table 4: Comparison of our proposed pipeline (with two different axiom settings) with the GAN baseline evaluated on the counterfactuals the GAN method produces in terms of accuracy, CF-MSE and worst subgroup CF-MSE (sg) as average of 5 runs. Best results are in bold.

| dataset | metric | LTN(acc) | LTN(acc+CF) | GAN |
|---------|--------|----------|-------------|-----|
| Adult | accuracy ↑ | $0.771 \pm 0.007$ | $\mathbf{0.775 \pm 0.005}$ | $0.758 \pm 0.006$ |
| | CF-MSE ↓ | $0.231 \pm 0.007$ | $\mathbf{0.200 \pm 0.007}$ | $0.208 \pm 0.025$ |
| COMPAS | accuracy ↑ | $0.672 \pm 0.015$ | $0.658 \pm 0.016$ | $\mathbf{0.680 \pm 0.001}$ |
| | CF-MSE ↓ | $0.259 \pm 0.016$ | $\mathbf{0.115 \pm 0.011}$ | $0.177 \pm 0.015$ |
| COMPAS(age) | accuracy ↑ | $0.653 \pm 0.013$ | $0.654 \pm 0.007$ | $\mathbf{0.668 \pm 0.006}$ |
| | CF-MSE ↓ | $0.321 \pm 0.011$ | $\mathbf{0.210 \pm 0.018}$ | $0.249 \pm 0.042$ |
| Lawschool | MSE ↓ | $0.235 \pm 0.002$ | $\mathbf{0.234 \pm 0.002}$ | $0.906 \pm 0.001$ |
| | CF-MSE ↓ | $0.019 \pm 0.015$ | $\mathbf{0.019 \pm 0.015}$ | $0.256 \pm 0.028$ |

Table 5: Comparison of our proposed pipeline with two different weight combinations for acc+CF axioms as well as for acc+CF_sg.

| dataset | metric | LTN (acc+CF) | | LTN (acc+CF_sg) | |
|---------|--------|:------------:|:---:|:---------------:|:---:|
| | | (1,1) | (2,1) | (1,1) | (2,1) |
| Adult | accuracy ↑ | $0.758 \pm 0.01$ | $0.772 \pm 0.006$ | $\mathbf{0.812 \pm 0.001}$ | $0.769 \pm 0.004$ |
| | CF-MSE ↓ | $0.065 \pm 0.002$ | $0.078 \pm 0.002$ | $\mathbf{0.055 \pm 0.001}$ | $0.066 \pm 0.001$ |
| | CF-MSE (sg) ↓ | $\mathbf{0.066 \pm 0.002}$ | $0.094 \pm 0.054$ | $0.084 \pm 0.044$ | $0.084 \pm 0.004$ |
| COMPAS | accuracy ↑ | $0.675 \pm 0.003$ | $0.674 \pm 0.013$ | $0.651 \pm 0.013$ | $\mathbf{0.683 \pm 0.009}$ |
| | CF-MSE ↓ | $0.047 \pm 0.004$ | $0.054 \pm 0.007$ | $\mathbf{0.045 \pm 0.006}$ | $0.052 \pm 0.002$ |
| | CF-MSE (sg) ↓ | $0.045 \pm 0.002$ | $0.057 \pm 0.012$ | $\mathbf{0.043 \pm 0.092}$ | $0.049 \pm 0.078$ |
| COMPAS(age) | accuracy ↑ | $0.654 \pm 0.016$ | $0.653 \pm 0.011$ | $\mathbf{0.658 \pm 0.012}$ | $0.655 \pm 0.018$ |
| | CF-MSE↓ | $0.079 \pm 0.016$ | $0.114 \pm 0.022$ | $0.075 \pm 0.016$ | $\mathbf{0.070 \pm 0.018}$ |
| | CF-MSE (sg)↓ | $\mathbf{0.131 \pm 0.021}$ | $0.269 \pm 0.091$ | $0.142 \pm 0.057$ | $0.143 \pm 0.078$ |
| Lawschool | MSE ↓ | $0.782 \pm 0.002$ | $0.773 \pm 0.018$ | $\mathbf{0.796 \pm 0.013}$ | $0.786 \pm 0.018$ |
| | CF-MSE ↓ | $0.003 \pm 0.000$ | $0.005 \pm 0.000$ | $\mathbf{0.001 \pm 0.000}$ | $0.002 \pm 0.000$ |
| | CF-MSE (sg) ↓ | $0.011 \pm 0.002$ | $0.011 \pm 0.001$ | $\mathbf{0.001 \pm 0.002}$ | $0.002 \pm 0.000$ |

### D.4. Influence of Axiom Weights

Our pipeline supports different weights for each group of axioms (accuracy, CF, CKE). This has direct influence on CF and accuracy as can be seen in Table 5. As a trend, accuracy improves, if higher weights are chosen for the accuracy axioms while CF decreases. However, this is not the case for all datasets, and we suggest here to try out different weight settings when applying our pipeline.

### D.5. Additional Knowledge Extraction

Integrating counterfactual fairness into a neurosymbolic framework poses several advantages. In particular, the satisfaction level to any logical query may be straightforwardly computed. This is of particular benefit in fairness-sensitive applications. Here we give two concrete applications for this.

Firstly, after training our pipeline, an individual can, for instance, run an existence query for the question: *is there a similar point in my subgroup which has a different outcome?* Concretely, for an individual data point $\hat{x}$ which is in subgroup $\mathcal{T}_C$ this query could be:

$$\exists x \in \mathcal{T}_C, x \neq \hat{x} : \neg D(x) = D(\hat{x}) \wedge ||x - \hat{x}||_2 < \tau \qquad (2)$$

Here, $\tau$ denotes the parameter for *similarity* and can be defined application-specific. In Table 6, we show examples to the output of this query in terms of an exemplary data point as well as the satisfaction level of the query for subgroups in the Adult dataset. We can see that exemplary data points differ in the features *age, education-level, marital-status, relationship and occupation.*

Secondly, the evaluation of CF can be flexibly queried for specific subgroups. This is especially interesting in applications where CF might not be relevant for all subgroups in the dataset as the application is specifically designed for one subgroup, e.g., giving out loans to teachers. Here, one can run auniversally-quantified query for this subgroup and evaluate if the model is CF with respect to the sensitive attribute. Overall, we conclude that a fair neurosymbolic method contributes to a wide range of additional knowledge extraction enhancing understanding of the underlying data and learning process.

Table 6: Satisfaction value (sat) and exemplary data point to the query *is there a similar point in my subgroup which has a different outcome?* Here, $\tau$ is equal to 3 for males and 5 for females.

| subgroup | sat | age | workclass | ed. | marital-status | occupation | relationship | h/w | nat.-country |
|---|---|---|---|---|---|---|---|---|---|
| white males | 0.217 | 23 | state-gov | 12 | never-married | adm-clerical | not-in-family | 39 | US |
|  |  | 24 | state-gov | 11 | married-civ-spouse | adm-clerical | husband | 39 | US |
| black males | 0.262 | 37 | private | 6 | married-civ-spouse | handlers-cleaners | husband | 39 | US |
|  |  | 36 | private | 8 | married-civ-spouse | machine-op-inspct | husband | 39 | US |
| white females | 0.270 | 33 | private | 8 | separated | adm-clerical | unmarried | 39 | US |
|  |  | 33 | private | 11 | divorced | adm-clerical | not-in-family | 39 | US |
| asian-pac-isl. females | 0.366 | 17 | private | 12 | never-married | exec-managerial | other-relative | 39 | Philippines |
|  |  | 16 | private | 12 | married-civ-spouse | exec-managerial | other-relative | 39 | China |

