# OpenReview forum: "A Neurosymbolic Approach to Counterfactual Fairness"
_nesyconf.org/NeSy/2025/Conference — NeSy 2025 Poster_

### Official Review · Reviewer_9ajW · 2025-04-04
**Very relevant paper for NeSy but lack of clarity**

**Rating:** 4
**Confidence:** 4

**Review:**

This paper presents a neurosymbolic pipeline for enforcing counterfactual fairness using Logic Tensor Networks (LTNs). The authors introduce logical axioms for enforcing accuracy and counterfactual fairness (CF), including subgroup-specific constraints, and propose a novel method of counterfactual knowledge extraction. The approach is evaluated on several benchmark datasets (Adult, COMPAS, COMPAS-age, and Lawschool) and compared against several state-of-the-art methods for counterfactual fairness.

The paper is clearly within the scope of the NeSy conference, as it effectively leverages LTNs to enforce counterfactual fairness. The research topic is both sensitive and highly relevant.
That said, I have some concerns regarding the clarity and novel contributions of the paper.

Clarity
The paper is somewhat difficult to follow due to its heavy reliance on the appendix. The reading flow is frequently interrupted by references to supplementary material, which makes it hard to track the core contributions and understand the methodology without switching back and forth. This structure might be better suited to a journal version, where details can be included in full, as and when needed.

Novelty and Contribution
In terms of novelty, it seems that the main contribution lies in the counterfactual knowledge extraction (CKE) component. If this is indeed the case, then this part should be highlighted more prominently and explained in greater detail in the main paper. At present, it risks being 'shadowed' by the broader LTN pipeline, which builds on existing work.

Significance of Results
The results appear to be reasonable and show improvement across datasets. However, it is not entirely clear whether the metric used to assess counterfactual fairness (CF-MSE) is a standard metric in the literature or an ad hoc one introduced by the authors. A discussion clarifying this would be helpful.

Also, the use of CNF (Causal Normalizing Flows) for counterfactual generation raises questions. Since the pipeline is agnostic to how counterfactuals are generated, why was CNF chosen specifically? A rationale or comparison would strengthen this aspect.

Minor Comments:
Section 2, Equation (1): Should the right-hand side of the equation read S = s′ instead of S = s?
Section 3: The brief introduction to LTNs could be moved to a related work or background section for better readability and organization.

**Anonymity:**

Remain anonymous

---

### Official Review · Reviewer_VqBP · 2025-04-07
**Counterfactual fairness with LTN executed simply and elegantly**

**Rating:** 7
**Confidence:** 3

**Review:**

The authors seek to address the problem of counterfactual fairness with LTN. To this end, they propose a first-order logic based axiomatization of the counterfactual constraints in LTN. The proposed axiomatization is intuitive, and the experimental results support its effectiveness.

STRENGTHS:
[Simplicity] The paper provides quite an intuitive solution for encoding counterfactual fairness constraints in LTN.

[Presentation] The paper is well-written, and all the ideas are clearly presented.

WEAKNESS:

[Soundness] The constraints that the paper seeks to axiomatize (equation 1) are probabilistic. In contrast, the authors model these constraints in a fuzzy logic framework. Although this seems to be the most natural and correct approach to achieve this in a model like LTN, I am not sure if this leads to a classifier that indeed obeys (eq 1) subject to having maximal accuracy. Because, to the best of my knowledge, LTN will maximize satisfaction of all the axioms. However, for a correct notion of counterfactual fairness, one may want to maximize accuracy, subject to equation 1 being satisfied. And these two notions of optimization are not necessarily equivalent. **It would be great if the effective equivalence of these two notions could be demonstrated theoretically.

**Anonymity:**

Remain anonymous

---

### Official Review · Reviewer_W9kQ · 2025-04-10
**A first solid attempt to integrate counterfactual fairness into ML systems**

**Rating:** 7
**Confidence:** 3

**Review:**

I apologise to the authors and the PC for delaying my review. I wanted to make sure I give the paper the adequate review time it deserved. I truly liked the paper.


This paper presents a novel approach to integrating counterfactual fairness (CF) into machine learning models using a neurosymbolic framework called Logic Tensor Networks (LTN). The authors aim to make machine learning models fairer by ensuring that protected attributes, like race or gender, do not bear casual influence on predicted outcomes. Authors integrate CF into the neural network learning process as logical constraints that are incorporated into the model during training. The paper demonstrates that their method improves fairness without significantly compromising accuracy, using experiments on real-world datasets. Additionally, the method can identify and address imbalances in fairness across different subgroups, making it a promising tool for creating more equitable AI systems. Overall, the method is quite flexible and generalisable, it is is agnostic to the underlying causal model and data generation technique. Thus, the paper advances the SOTA in neurosymbolic fairness approaches and is the first one to integrate counterfactual fairness, a very important issue in itself, to a neurosymbolic framework.

The paper is well structured and clearly written. The three types of axioms used do make sense and are encoded straightforwardly into FOL. As for the evaluation, it focuses on three main aspects: overall and subgroup counterfactual fairness, comparison with other approaches, and the effectiveness of counterfactual knowledge extraction. The experiments are conducted on multiple real-world datasets (Adult, COMPAS, and Lawschool), which adds credibility to the results by demonstrating applicability across different contexts.

However, there are a few potential weaknesses or areas for improvement in the experiment design and results:
1. While the datasets used are well-known, they primarily focus on binary and categorical sensitive attributes. Including more diverse datasets with different types of sensitive attributes or more complex data structures could strengthen the generalizability of the findings.
2. The paper compares its method against a few baselines, but the choice of baselines and the details of their implementation could be more thoroughly justified.
3. The paper does not extensively discuss the computational efficiency or scalability of the proposed method, which could be a concern for large-scale applications.
4. It only tries out the integration with the simplest MLP architecture while SOTA for machine learning lies elsewhere.

Overall, the paper presents a novel integration of counterfactual fairness into neurosymbolic AI, which is a relatively unexplored area. More so, The focus on subgroup counterfactual fairness is a valuable addition, addressing fairness at a more granular level. The methodology for automatic extraction of fairness constraints from counterfactual explanations is also an advancement, potentially reducing manual intervention. The proposed method seems quite versatile and in the limited settings of current evals seems like a nice tradeoff between preserving accuracy and enforcing fairness.
The main criticism from my side, as stated above, is around how generalisable and scalable the system could be.
Addition aspects that could benefit the paper could be:
- Can you elaborate on how the choice of axiom weights impacts both fairness and accuracy across different datasets?
- How does your method scale with larger datasets, and what are the computational requirements?
- Are there any specific types of datasets or applications where your method might not perform well?

Despite some areas for improvement, particularly in terms of dataset diversity and baseline comparisons, the work is innovative and well-executed. I recommend acceptance with minor revisions to address the concerns raised.

**Anonymity:**

Remain anonymous